# Retinoic Acid Potentiates the Therapeutic Efficiency of Bone Marrow-Derived Mesenchymal Stem Cells (BM-MSCs) against Cisplatin-Induced Hepatotoxicity in Rats

**Maha M. Azzam** [1], **Abdelaziz M. Hussein** [2,*], **Basma H. Marghani** [1,3], **Nashwa M. Barakat** [4], **Mohsen M. M. Khedr** [5] and **Nabil Abu Heakel** [1]

[1] Department of Physiology, Mansoura Faculty of Veterinary Medicine, Mansoura University, Mansoura 35516, Egypt
[2] Department of Medical Physiology, Mansoura Faculty of Medicine, Mansoura University, Mansoura 35516, Egypt
[3] Department of Biochemistry, Physiology and Pharmacology, Faculty of Veterinary Medicine, King Salman International University, South of Sinaa 46612, Egypt
[4] Urology & Nephrology Center, Mansoura University, Mansoura 35516, Egypt
[5] Department of Chemistry, Faculty of Science, Port Said University, Port Said 42521, Egypt
* Correspondence: zizomenna28@yahoo.com or menhag@mans.edu.eg; Tel.: +20-10-0242-1140

**Abstract:** (1) Background: Hepatotoxicity is a common health problem, and oxidative stress plays a crucial role in its underlying mechanisms. We inspected the possible effect of retinoic acid (RA) in the potentiation of hepatoprotective effect of bone marrow mesenchymal stem cells (BM-MSCs) against Cisplatin (Cis)-induced hepatotoxicity. (2) Methods: 60 male Sprague Dawley rats (SD) were separated randomly and designated to six main equal groups as follows: (1) Control group, (2) Cis group (rats got Cis 7 mg/Kg i.p.), (3) Cis + vehicle group (as group 2, but rats received the (vehicle) culture media of BM-MSCs), (4) Cis as in group 2 + BM-MSCs ($1 \times 10^6$), (5) Cis as for group 2 + RA 1 mg/Kg i.p., and (6) Cis and BM-MSCs as for group 3 + RA as for group 4. Liver injury was assessed by measuring liver enzymes (ALT, AST), while liver toxicity was evaluated by histopathological examination. Apoptotic marker caspase-3 protein was detected immunohistochemically. Real time PCR was performed to detect NADPH oxidase and TNF-$\alpha$ at transcription levels. Oxidative stress was investigated by colorimetric measurement of MDA, GSH and catalase. (3) Results: Contrary to the Cis group ($p < 0.05$), BM-MSCs/RA supplementation resulted in a substantial decrease in serum levels of hepatic impairment indicators such as ALT, AST and oxidative stress markers such as MDA, as well as an increase in hepatic GSH, Catalase, and a decrease in expression of TNF-$\alpha$ and downregulation of NADPH oxidase. The improvement after therapy with BM-MSCs/RA was confirmed by histopathological examination. Moreover, the downregulation of caspase-3 in liver tissue after BM-MSCs/RA treatment was validated by immunohistochemistry investigation. (4) Conclusions: BM-MSCs and RA attenuated Cis induced hepatotoxicity through downregulation of oxidative stress resulted in modulation of anti-inflammatory TNF-$\alpha$ and apoptosis caspase-3 indicating a promising role in hepatotoxicity.

**Keywords:** cisplatin hepatotoxicity; BM-MSCs; retinoic acid; oxidative stress; NADPH oxidase; apoptosis

## 1. Introduction

Liver disease is regarded as one of the world's most dangerous health issues, with a high mortality rate [1]. Cisplatin (Cis) is a powerful anticancer chemotherapeutic agent; however, it has a toxic effect on various tissues, e.g., ototoxicity (75–100%), nephrotoxicity (72%), hepatotoxicity and cardiotoxicity (6%) [2]. Several mechanisms

have clarified the pathogenesis of Cis-induced toxicity such as oxidative stress, apoptosis and inflammatory mediators [2–5]. Pro-inflammatory cytokines such as tumor necrosis factor-$\alpha$ (TNF-$\alpha$) and interleukin-6 (IL-6) are generated as a consequence of the Cis excessive production of reactive oxygen species, which accelerate the process of cytotoxicity and apoptotic cell death [6,7]. ROS results in the damage of the cell membranes via lipid peroxidation and activating neutrophils which consequently increases a membrane-bound NADPH oxidase enzyme [4]. It has been demonstrated that inhibiting NADPH oxidase can reduce oxidative stress-related damage [8,9]. Moreover, Cis activates the apoptotic cell death by activation of p53 and caspase-3 and suppression of anti-apoptotic Bcl2 proteins in tissues [2].

Mesenchymal stem cells (MSCs) are hopeful sources for cell-based therapy and regenerative medicine. The use of stem cells as a cell replacement therapy for liver toxicity could help in solving the obstacle of the limited number of liver donors [10]. Administration of MSCs could be beneficial in regenerative medicine due to its migration potential to the damaged tissue or organ [11,12]. Bone marrow (BM)-MSCs participate in tissue repairing because of their abilities of differentiation into tissue-specific cell types [13,14], by the production of trophic factors that minimize the tissue damage and help in tissue repair [15] or by activating the immune system [16]. Moreover, a previous study demonstrated that anti-oxidant and anti-apoptotic effects for BM-MSCs in kidney tissues, so we hypothesize that the BM-MSCs could target the pathways of Cis-induced hepatotoxicity [17].

Retinoids are powerful regulators of several biological processes in the cells, e.g., cell proliferation, differentiation and apoptosis [17–19]. Several clinical trials demonstrated that retinoic acid (RA), one of the Retinoids, is a good differentiating factor [20]. Moreover, previous studies reported the expression of retinoid receptors in hepatocytes [21]. Moreover, it has been demonstrated that hematopoietic and non-hematopoietic stem cells can differentiate into hepatocyte-like cells [22,23]. A study by Wang and his colleagues demonstrated that RA helps the liver progenitor cells differentiate into hepatocyte-like cells [24]. Hence, we hypothesized that the addition of RA to BM-MSCs could improve their therapeutic efficacy against Cis-induced hepatotoxicity.

## 2. Material and Methods

### 2.1. Drugs and Chemicals

Cis and RA were obtained (Sigma Aldrich Chemical Co., St. Louis, MO, USA). Cat No. (F4135) fetal bovine serum, qualified, heat-inactivated (Sigma Aldrich Chemical Co., St. Louis, MO, USA). (Cat. No. (M0894) Minimum essential medium ($\alpha$-MEM) (Sigma Aldrich Chemical Co., St. Louis, MO, USA). (Cat. No. (217004) RNeasy Plus Mini Kit (Qiagen, Hilden, Germany). Cat. No. (330404) cDNA an RT2 First Strand Kit (Qiagen, Hilden, Germany). Cat. No. (204141) SYBR Green PCR Master Mix (Qiagen, Hilden, Germany).

### 2.2. Experimental Animals and Study Design

Sixty male Sprague Dawley (SD) rats (200 ± 25 g) ranging in age from 2–3 months were procured from Mansoura University's Faculty of Medicine in Egypt's Medical Experimental Research Center. The animals were kept in approved animal cages at Mansoura University's Faculty of Veterinary Medicine, Egypt, under conventional environmental circumstances, such as a room temperature of 22 °C, a 12 h light/dark cycle, and free access to food and water throughout the study. Before starting the experiment, the rats were kept for 7 days to acclimate to the laboratory environment. All experimental animals used in this study had been cared for according to the National Institute of Health guide for the care and use of laboratory animals, as well as Research Ethics Committee, Faculty of Veterinary Medicine, Mansoura University, Mansoura, Egypt), Code No. ph.D/17.

Five groups of ten rats each were randomly subdivided into: (1) Control group; normal rats were given 0.5 mL of saline intraperitoneally (i.p.). (2) Cis group: solitary dose of Cis (7 mg/kg) dissolved in 0.5 mL saline was given to rats (i.p.) to induce liver injury [25]. (3) Cis + vehicle group: one day following Cis injection, rats were given a solitary dose of 0.5 mL culture media (vehicle) in the tail vein. (4) Cis + BM-MSCs group: one day following Cis injection, rats were given solitary dose of 0.5 mL culture media contain BM-MSCs ($1 \times 10^6$) in the tail vein. (5) Cis + RA group: one day following Cis injection, rats were given a single dose of RA (1 mg/kg) in the tail vein. (6) Cis + BM-MSCs + RA: rats treated with BM-MSCs-pretreated with RA in the same previous doses. On days 3, 7 and 11 of the experiment, rats were anesthetized with diethyl ether; then, capillary hematocrit tubes were used to collect blood samples from the retro-orbital plexus. Blood samples were obtained and centrifuged for 10 min at 3000 rpm after being kept at room temperature for 15 min. Serum was isolated and stored at −80 °C for biochemical analysis of the liver enzymes alanine aminotransferase (ALT) and aspartate aminotransferase (AST). The liver of each rat was collected and washed using normal physiological saline for removing any remaining blood before being processed. Phosphate-buffered saline was used to homogenize 1 g of liver tissue (0.01 M sodium phosphate buffer, pH 7.4, containing 0.14 M NaCl). After 15 min of centrifugation at 4000 rpm, Until the investigation, the supernatant was collected and frozen at −80 °C.

### 2.3. Isolation, Expansion and Characterization of Rat BM-MSCs

Huang et al. described the isolation of rat BM-MSCs from SD rats [26]. Four-week-old rats were terminated by cervical dislocation, and the hindlimbs femur and tibias after removal of the muscle tissues were soaked for two minutes in a 70% (*v/v*) solution and then rinsed twice with phosphate-buffered saline (PBS) containing 1% penicillin/ streptomycin. The bones were placed in a 100 mm aseptic culture dish containing 10 mL of complete $\alpha$-MEM (Sigma Aldrich, St. Louis, MO, USA) added to it 15% fetal bovine serum (Sigma Aldrich, St. Louis, MO, USA). After being shifted into the biosafety cabinet, to remove impurities, the dish was washed twice, and the two ends just below the marrow cavity's end were excised. A needle of 23-gauge was pushed into the bone cavity, and a complete $\alpha$-MEM medium was used to drain the marrow out gently. The marrow-containing medium was then cultured in a 25 cm² cell culture flask (Corning Inc, Corning, NY, USA) and incubated at 37 °C in a 5% $CO_2$ incubator. After 24–72 h, the medium was changed, and non-adherent cells were eliminated. When, the culture reached 70–90% confluence, the cells were digested with 2.5 mL of 0.25% trypsin at 37 °C for 2 min and cultured in a 75 cm² cell culture flask (Corning Inc, Corning, NY, USA) until passage 3. The cultured cells were characterized using a FACS caliber flow cytometer (Beckman, Fullerton, CA, USA) at the children's hospital at Mansoura University for CD44, 106, 29, 34, 14 and 45. Flow-cytomtery revealed that these cells expressed high levels of CD44, CD106 and CD29 but negligible levels of hematopoietic CDs (CD34, CD14 and CD45) as illustrated in Figure 1.

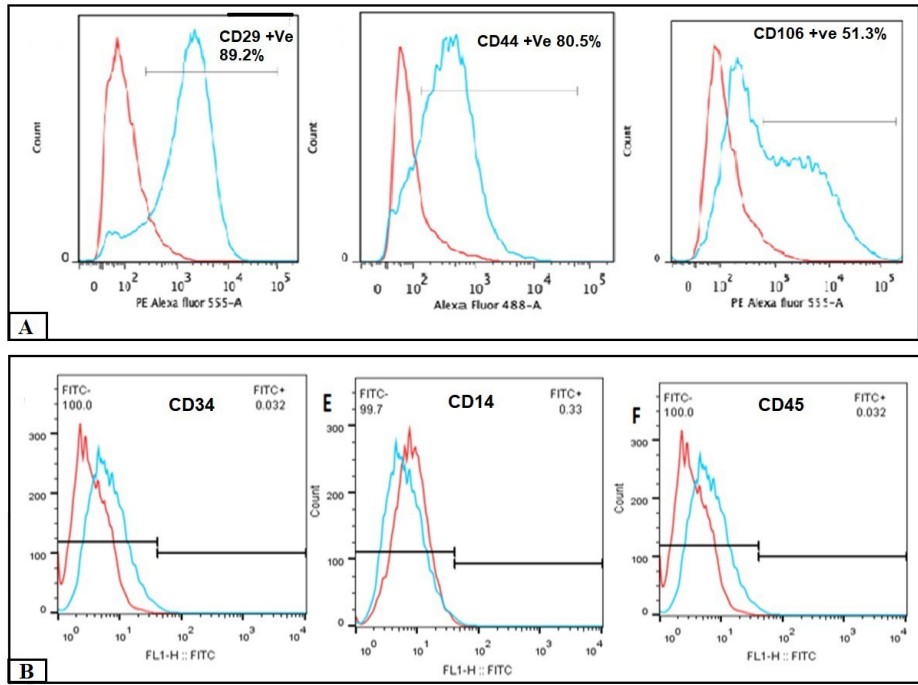

**Figure 1.** BM-MSCs characterization. Flow cytometry histograms for BM-MSCs showing positivity for CD29, CD44 and CD106 (**A**) and negativity for CD34, CD14 and CD45 (**B**).

The levels of ALT and AST were determined by an automatic biochemical analyzer (Beckman Coulter, Shanghai, China), according to Reitman and Frankel's colorimetric approach [27], using commercially available kits SPINREACT (quantitative determination of ALT and AST) (Ctra. Santa Coloma, Sant Esteve De Bas, Girona, Spain). One milliliter of rat serum was drawn into a test tube and incubated for 10 min in a water bath at (40 °C). After adding 0.2 mL of serum, the contents were mixed and incubated for 60 min for AST or half an hour for ALT. One ml of 2,4-dinitrophenylhydrazine reagent was added immediately, and after 20 min at room temperature, 10 mL of 0.4 N sodium hydroxide was added into the tube, and the contents were mixed. After 30 min, the optical density was measured at 505 nm.

### 2.4. Evaluation of Oxidative Stress Marker

Rat Malondialdehyde (MDA), reduced glutathione (GSH) and catalase (CAT) activity assay kits (Bio-Diagnostic, Giza, Egypt) were used. Rat MDA kit (Bio-Diagnostic, Giza, Egypt) was used to assess the MDA levels in liver homogenate, which is based on thiobarbituric acid reacting with MDA to form thiobarbituric acid reactive species (pink-colored product), which were calorimetrically measured at 534 nm and relative to MDA content. The MDA levels in liver tissue were expressed as nmol/g tissue [28].

### 2.5. Evaluation of Antioxidant Activities

Determination of GSH concentration in liver homogenate was performed by rat GSH kit (Bio-Diagnostic, Giza, Egypt), based on the reaction of 5,5 dithiobis 2-nitrobenzoic acid (DTNB) with glutathione, where a relatively stable yellow color developed and measured spectrophotometrically at 412 nm [29]. The concentration of CAT in liver homogenate was assayed by rat CAT kit (Bio-Diagnostic, Giza, Egypt), based on CAT reacted with a known quantity of hydrogen peroxide ($H_2O_2$), which was calorimetrically measured at 510 nm [30].

### 2.6. Gene Expression by Real-Time Polymerase Chain Reaction

RNeasy Plus Mini Kit (Qiagen GmbH, Hilden, Germany) was used to extract total RNA from rat liver. One µg of total RNA was used and converted to cDNA using an RT2 First Strand Kit (Qiagen Sciences, Germantown, MD, USA). Primers for the reference gene (GAPDH), TNF alpha, and NADPH oxidase were designed with NCBI to assist as shown in Table 1 and manufactured by VIVANTS (Selangor, Malaysia). In each well, amplifications were performed by 25 µL reaction volume (12.5 µL of 2X SYBR Green Master Mix (Qiagen, Hilden, Germany), 1 µL of cDNA template, 1 µL of gene primer, and 10.5 µL of nuclease-free water). The reaction of amplification was created in a real-time thermal cycler (CFX96 Real-Time System, Bio-Rad, Hercules, CA, USA) and programmed according to the manufacturer's instructions. The quantity of the targeted gene expression was relative to those obtained for normal rat liver.

**Table 1.** Primer sequence and annealing temperature of investigated genes.

| | Accession No. | Primer Sequence | Product Length | Annealing Temperature |
|---|---|---|---|---|
| GAPDH | NM_053524.1 | F: TGCCACTCAGAAGACTGTGG | 85 | 59 |
| | | R: GGATGCAGGGATGATGTTCT | | |
| TNF-$\alpha$ | NM_012675.3 | F: TCTTCAAGGGACAAGGCTGC | 104 | 60 |
| | | R: CTTGATGGCAGAGAGGAGGC | | |
| NADPH | NM_017008.4 | F: TGTTGGGCCTAGGATTGTGT | 119 | 60 |
| | | R: CTTCTGTGATCCGCGAAGGT | | |

### 2.7. Histopathology and Caspase-3, p53, Bcl2 and IL1beta Immunohistochemistry

For histopathological analysis and immunohistochemical staining for the apoptotic marker (Caspase-3), liver tissue was fixed in 10% neutral buffered formalin. A light microscope (BH-2; Olympus, Tokyo, Japan) was used to examine five-µm thick sections of paraffin-embedded liver specimens stained with hematoxylin and eosin (H&E). The severity of liver impairment was assessed semi-quantitatively as follows: necrosis, inflammatory cell infiltration, fatty change, and focal congestion and hemorrhage. For each criterion, the scores of hepatic lesions were assigned as follows: absent (0), slight (1), moderate (2), and severe (3) with maximum score 12 [31].

Deparaffinized and rehydrated liver sections were then incubated with citrate buffer solution (pH 6.0), boiled in a pressure cooker at 94 °C for 20 min, then cooled for 20 min at room temperature. After that, the sections were washed in phosphate-buffered saline (PBS). Endogenous peroxidase was inhibited by incubating cells for 15 min at room temperature in a 3% hydrogen peroxide solution before washing with PBS. The sections were then treated with protein block and incubated for 60 min with the primary antibodies, then washed with PBS and incubated for 20 min with biotinylated goat anti polyvalent and streptavidin Px at room temperature. For 10 min, the staining was completed with chromogen + substrate. The slides were counterstained for 1 min with Mayer's hematoxylin, rinsed in tap water and dehydrated. Caspase-3 protein positive signal was brown granular mass, which could be used to trace this protein. A total of ten fields were chosen randomly and examined at a magnification of 20. A Leica DFC280 camera with Labomed Cxl light microscope and a Leica Q Win Image Analysis system were used to examine all sections (Leica Micros Imaging Solutions Ltd., Cambridge, UK). The primary antibodies were polyclonal rabbit antibodies and purchased from Thermo Fisher Scientific, Inc., Waltham, MA, USA (for caspase-3) and Santa Cruz Biotechnology Inc, Santa Cruz, CA, USA (for p53, Bcl2 and IL1beta). To assess the expression of caspase-3 positive signal, a semiquantitative score from 0 to 3 was assigned: (0) when negative (no expression), (1) minimal expression (involving ≤25% of the liver), (2) moderate expression (involving 25–50% of the liver) and (3) severe expression (involving

≥50% of the liver) in 10 high power fields [32]. Moreover, scoring of the expression of p53, Bcl2 and Il1beta in liver tissues was done as follows, we examined the positive expression in 9 high power fields (HPF), no expression = 0, 1–2 field exhibited immunoreactive staining/9 HPF = 1, 3–5 field exhibited immunoreactive staining/9 HPF = 2 and 6–9 field exhibited immunoreactive staining/9 HPF = 3.

### 2.8. Statistical Analysis

All the data obtained from the experiment were expressed as means ± SEM. Statistical analysis of data was carried out by the Statistical Package for the Social Sciences (SPSS Statistics 17.0) (SPSS, Chicago, IL, USA) using the one-way ANOVA with Tukey post hoc test for testing the significant differences between variables. Results were considered significant only at the level of ($p < 0.05$). Total histopathological scores from each group at each sacrifice were analyzed using Kruskal–Wallis test followed by Tukey post hoc and to compare different means. $p < 0.05$ was considered to be significant.

## 3. Results

### 3.1. Effect of BM-MSCs and RA and Combination of Both on Liver Functions in Cis-Induced Hepatotoxicity

The results of liver enzymes (ALT and AST) showed significant elevation in their levels in Cis and Cis + vehicle groups compared to control group ($p < 0.05$) at different studied time intervals. On the other hand, Cis + MSCs group, Cis + RA group and Cis + MSCs + RA group showed significant reduction in the plasma levels of the liver enzymes compared to Cis and Cis + vehicle groups ($p < 0.05$) at different time intervals. Moreover Cis + RA+ MSCs group showed more significant attenuation in plasma levels of ALT and AST compared to Cis + RA and Cis + MSCs group ($p < 0.05$) at different time intervals (Figure 2).

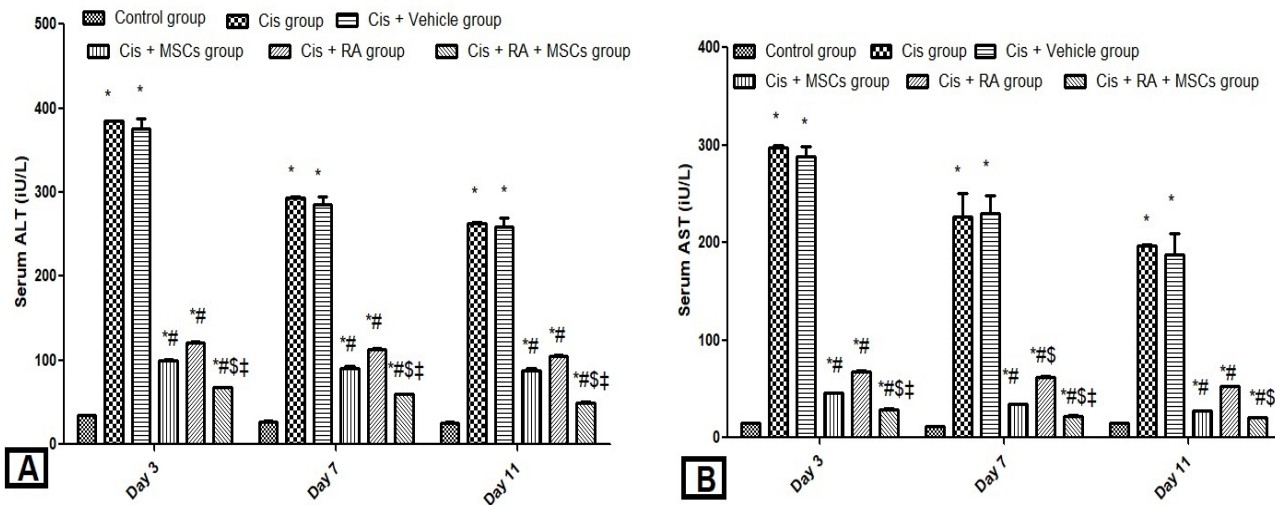

**Figure 2.** Serum levels of liver enzymes: ALT (**A**) and AST (**B**) (iU/L) in different studied groups. All data are expressed as Mean ± SD. One-way ANOVA with Tukey post hoc test. * Significant vs. control group, # significant vs. Cis group, ‡significant vs. Cis + BM-MSCs group and $ significant vs. Cis + RA group. ALT = alanine transaminase and AST = aspartate transaminase, Cis = cisplatin, RA = retinoic acid and BM-MSCs = bone marrow derived mesenchymal stem cells. $p < 0.05$ is considered significant.

### 3.2. Effect of BM-MSCs and RA and Combination of Both on Oxidative Stress Markers (GSH, CAT and MDA) in Cis-Induced Hepatotoxicity

The concentrations of GSH and the activity of CAT were significantly reduced in liver tissues in Cis and Cis + vehicle groups compared to control group at different time intervals ($p < 0.05$). On the other hand, they were significantly increased in Cis + RA, Cis + MSCs and Cis + RA + MSCs groups compared with Cis and Cis + vehicle groups ($p < 0.05$). Moreover, GSH concentration and CAT activity in liver tissues were significantly increased in Cis +RA + MSCs group compared to Cis + RA and Cis + MSCs groups ($p < 0.05$). On the other hand, the concentration of MDA in liver tissues was significantly higher in Cis and Cis + vehicle groups compared with control group ($p < 0.05$), while its concentration was significantly attenuated in all treated groups compared to Cis and Cis + vehicle groups ($p < 0.05$). In addition, MDA concentration in liver tissues was significantly attenuated in Cis + RA + MSCs group compared to Cis + RA and Cis + MSCs groups ($p < 0.05$) (Figure 3).

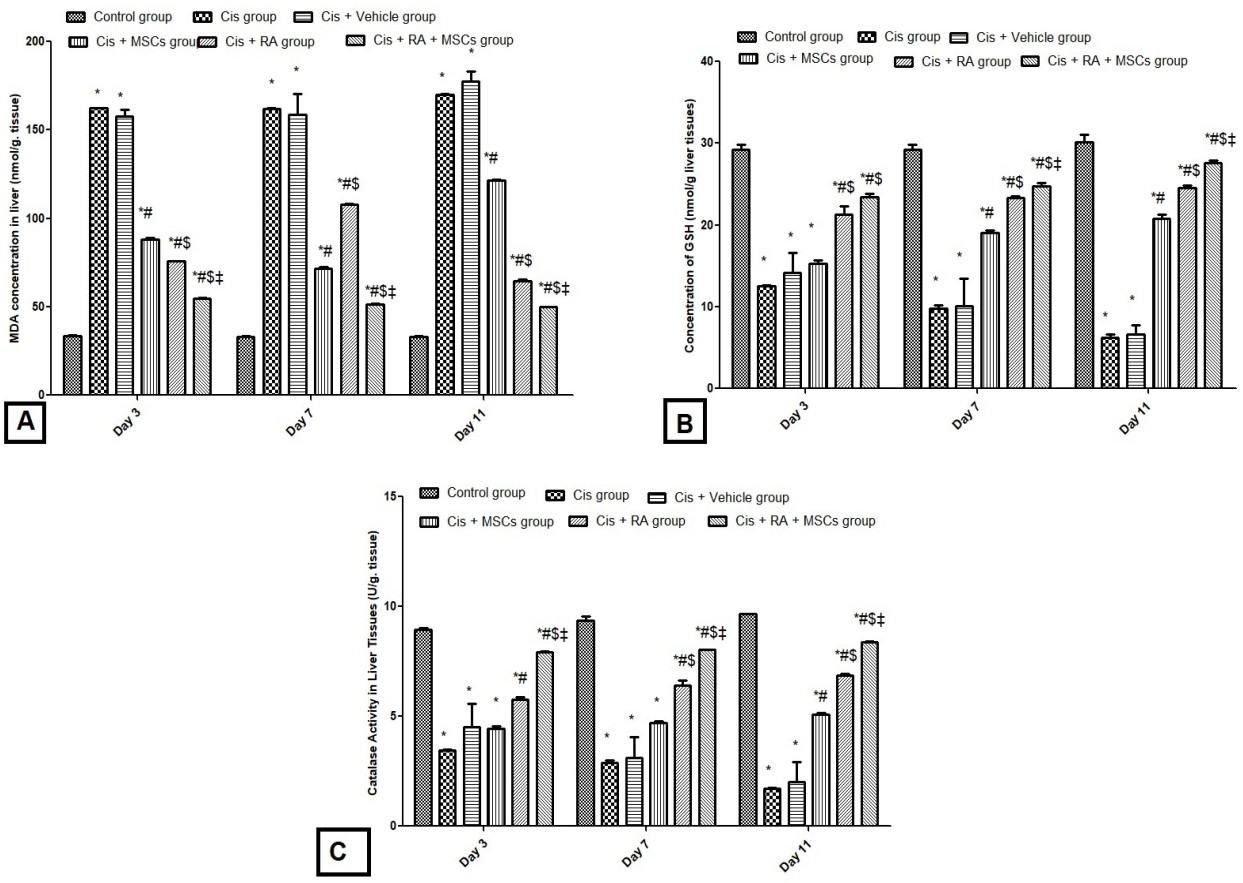

**Figure 3.** Markers of oxidative stress including MDA (**A**), GSH (**B**) and catalase (**C**) in different studied groups. All data are expressed as mean ± SD. One-way ANOVA with Tukey pos thoc test. * Significant vs. control group, # significant vs. Cis group, ‡ significant vs. Cis + BM-MSCs group and $ significant vs. Cis + RA group. MDA = malondialdehyde and GSH = glutathione, CAT = catalase, Cis = cisplatin, RA = retinoic acid and BM-MSCs = bone marrow derived mesenchymal stem cells. $p < 0.05$ is considered significant.

### 3.3. Effect of BM-MSCs and RA and Combination of Both on NADPH Oxidase and TNF-α Expression in Cis-Induced Hepatotoxicity

The expressions of the NADPH oxidase gene in liver tissues were significantly increased in Cis and Cis + vehicle groups compared to control group ($p < 0.05$). Moreover, Cis + RA + MSCs showed significant attenuation in NADPH oxidase in liver tissues compared to Cis group and the other treated group ($p < 0.05$) (Figure 4A). On the other hand, the expression of TNF-α was significantly higher in Cis group, Cis + vehicle groups, Cis + RA group and Cis + BM-MSCs group compared to control group ($p < 0.05$) at different time intervals. Moreover, Cis + RA+ BM-MSCs group was the only group that showed significant attenuation in TNF-α expression compared to Cis group and other treated groups at different time intervals ($p < 0.05$) (Figure 4B).

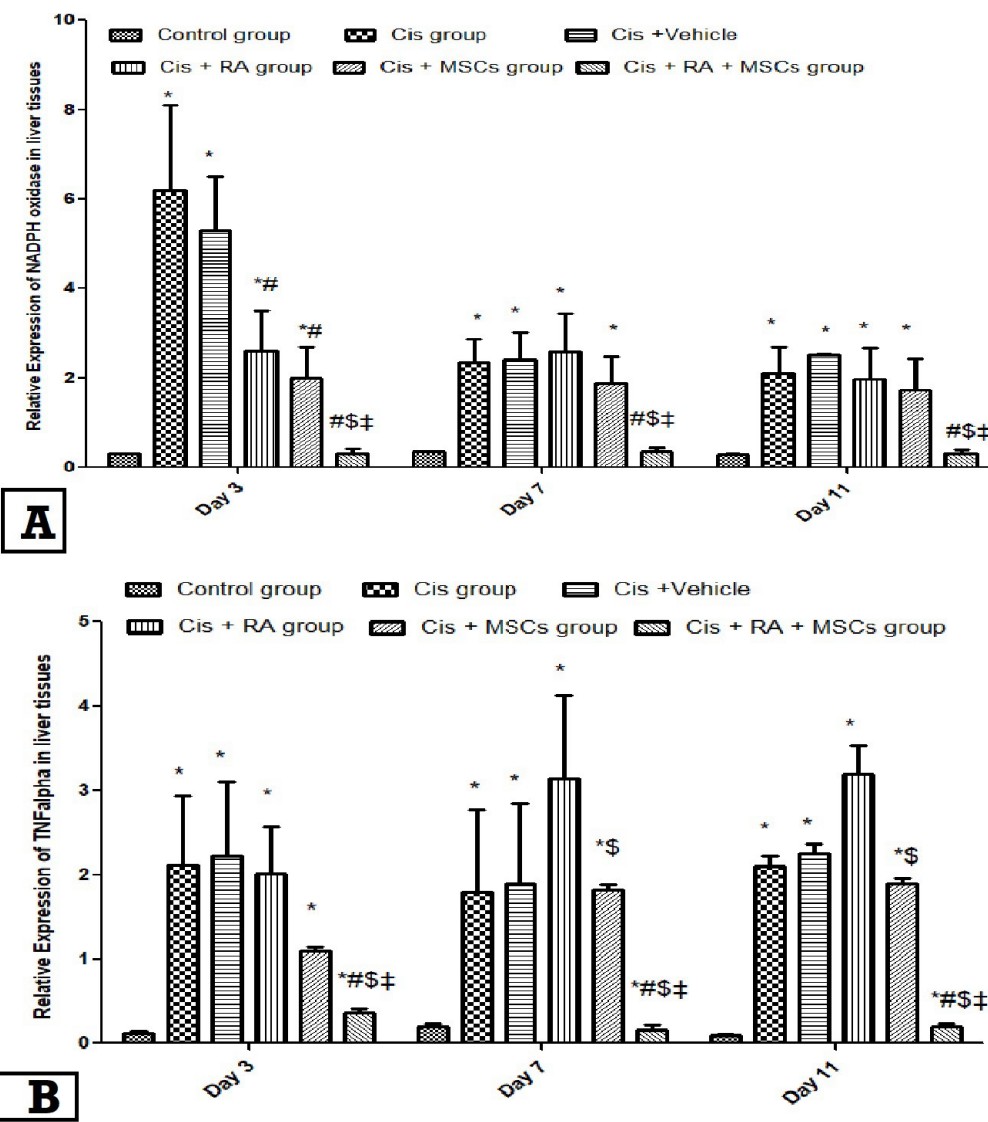

**Figure 4.** Relative gene expression of (**A**) TNF-alpha and (**B**) NADPH oxidase genes at the level of mRNA. * Significant vs. control group, # significant vs. Cis group, $ significant vs. Cis + MSCs group and ‡ significant vs. RA group.

*3.4. Effect of BM-MSCs and RA and Combination of Both on Liver Morphology in Cis-Induced Hepatotoxicity*

Liver histological damage scores showed significant increase in the severity of liver injury in Cis group compared to control and vehicle groups at 3 times of sacrifice ($p < 0.05$), while all treated groups showed significant attenuation in damage scores compared to Cis group ($p < 0.05$), and the more significant attenuation was noted in Cis + RA + MSCs group (Figure 5A). Liver specimens from normal control group showed no cellular deformities with normal liver architecture including bile ducts, endothelial cells and vessels in portal region (Figure 5B,C). On the other hand, liver specimens from Cis group showed disrupted hepatic cords, interstitial edema (yellow arrows), inflammatory cell infiltrate (red arrows) and diffuse hydropic degeneration in hepatocytes accompanied with focal fatty changes (blue arrows) (Figure 5D,E). Vehicle group shows the same histopathological changes (Figure 5F). Moreover, the liver specimens from rats treated with MSCs, RA or a combination of both showed less inflammatory cell infiltrates, less apoptotic cells with prominent nuclei and mitotic figures (Figure 5G–J).

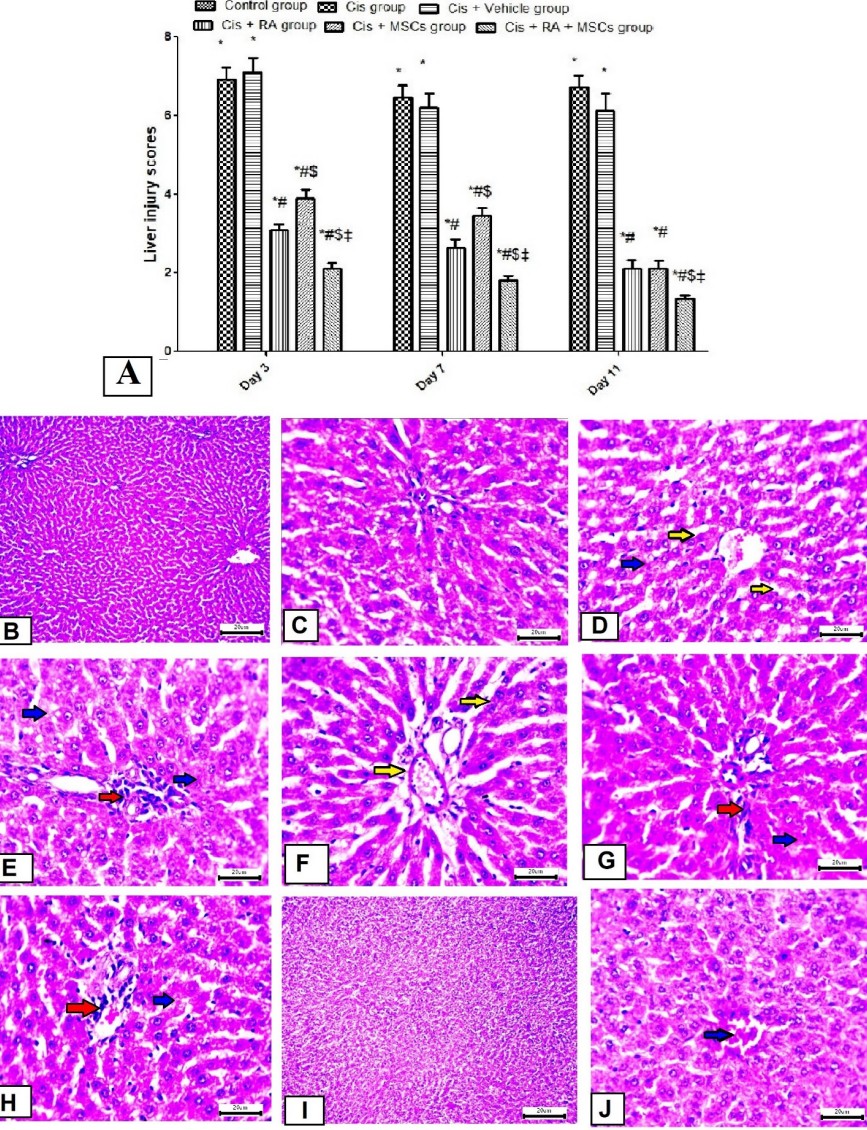

**Figure 5**. Histopathological examination of liver specimens. The score of liver damage at different time intervals (3 days, 7 days and 11 days) showed significant rise in damage score in Cis and

vehicle groups and significant attenuation in all treated groups (Cis + RA, Cis + BM-MSCs and Cis + RA + BM-MSCs groups) (**A**). Microscopic images from control group show normal radial arrangement of hepatic cords around central veins with normal portal areas and sinusoids at 11 days from start of experiment ((**B**) 100× and (**C**) 400×). Liver sections from Cis group show disrupted hepatic cords, interstitial edema (yellow arrow) and inflammation (red arrows), diffuse hydropic degeneration in hepatocytes (blue arrows) and occluded sinusoids ((**D,E**) 400×). Moreover, liver specimens from vehicle group shows the same morphological findings ((**F**), 400×). Liver specimens from Cis + BM-MSCs group ((**G**) 400×), Cis + RA ((**H**) 400×) and Cis + RA + BM-MSCs ((**I**) 100× and (**J**) 400×) at 11 days show improved liver pictures with milder hydropic degeneration in hepatocytes detected in some sections than in untreated Cis group after 11 days. * Significant vs. control group, # significant vs. Cis group, $ significant vs. Cis + MSCs group and ‡ significant vs. RA group.

### 3.5. Effect of BM-MSCs and RA and Combination of Both on Apoptotic Markers (Caspase-3, p53 and Bcl2) in Cis-Induced Hepatotoxicity

Immunohistochemical examination for caspase-3 and p53 showed a significant increase in their expression levels in Cis group when compared with the control group ($p < 0.05$), while other groups showed significant decrease in caspase-3 and p53, with the maximum significant attenuation in Cis + RA + MSCs group compared with Cis group ($p < 0.05$) (Figures 6A and 7A). Immunopositivity for caspase-3 appeared as cytoplasmic brown staining in liver in different treated groups at different times (Figure 6B–H). Moreover, immunopositivity for p53 appeared as cytoplasmic brown staining in hepatocytes at different times (days 3,7 and 11) in different groups (Cis, RA + Cis, MSCs + Cis and RA + MSCs + Cis groups) (Figure 7B–H).

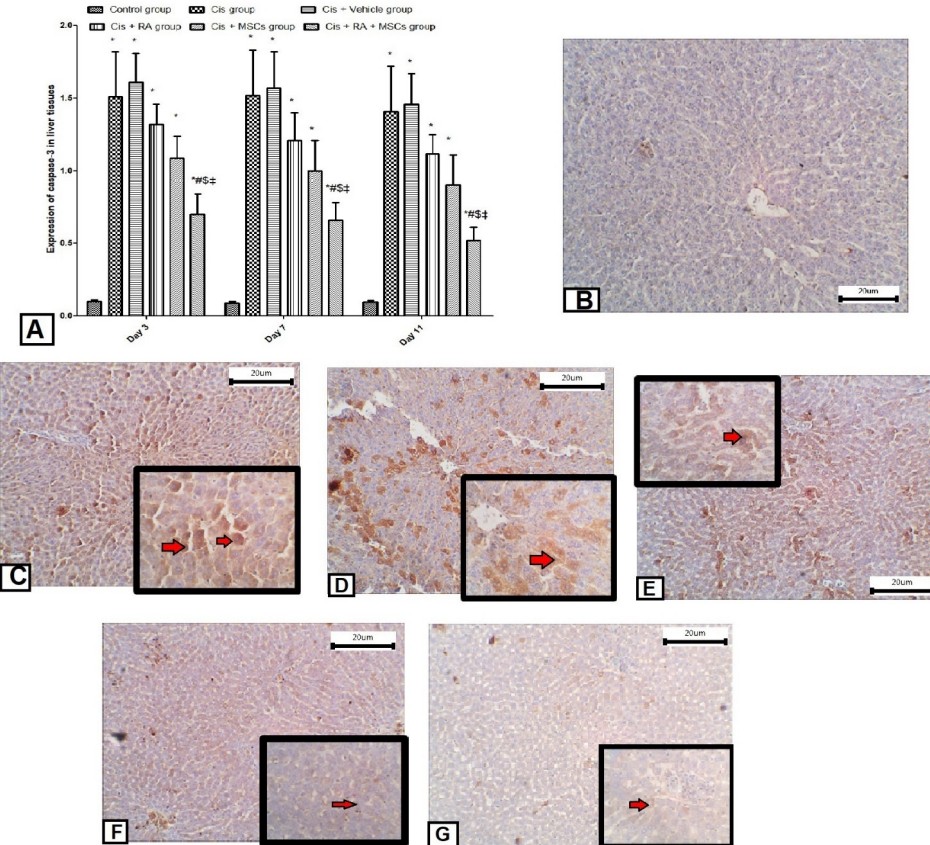

**Figure 6**. Immunohistopathological examination for the expression of caspase-3 in liver specimens. The score of caspase-3 expression in liver at different time intervals (3 days, 7 days and 11 days)

showed significant rise in its expression in Cis group with significant attenuation Cis + RA + BM-MSCs group (**A**). Microscopic pictures of immunostained hepatic sections stained with caspase-3 from rats groups sacrificed at 11 days showing negative staining in control group ((**B**) 100×), increased positive brown expression of caspase-3 (red arrows) in hepatocytes in Cis group group ((**C**) 100× with high power 400× side box), Cis + vehicle group ((**D**) 100× with high power 400× side box), Cis + RA group ((**E**) 100× with high power 400× side box) and weak expression in BM-MSCs group ((**F**) 100× with high power 400× side box), and Cis + BM-MSCs + RA group ((**G**) 100× with high power 400× side box). * Significant vs. Control group, # significant vs. Cis group, $ significant vs. Cis + MSCs group and ‡ significant vs. RA group.

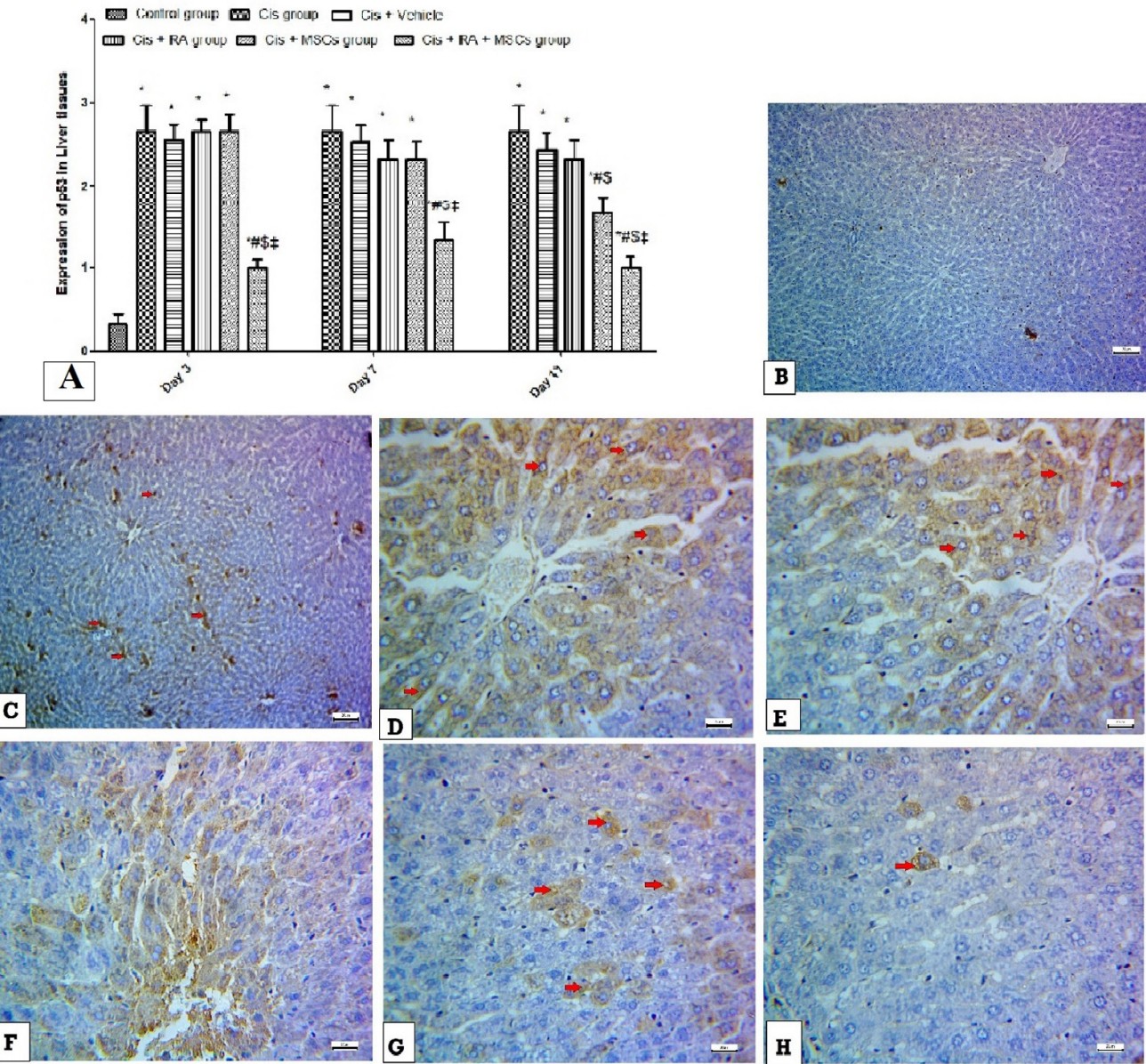

**Figure 7.** Immunohistopathological examination for the expression of p53 in liver specimens. The score of p53 expression in liver at different time intervals (3 days, 7 days and 11 days) showed significant rise in its expression in Cis group and significant attenuation in all treated groups (Cis + RA, Cis + BM-MSCs and Cis + RA + BM-MSCs groups) (**A**). Microscopic pictures of immunostained hepatic sections against p53 from rats groups sacrificed at day 11 showing negative staining in control group ((**B**) 100×), increased positive brown expression of p53 (red arrows) in hepatocytes in

Cis group group at day 11 ((**C**) 100× and (**D**) 400×) and vehicle group (**E**) in contrast to treated groups where positive cytoplasmic brown expression (red arrows) in hepatocytes is moderate in group treated with RA group ((**F**) 400×), MSCs group ((**G**) 400×), to minimal expression in group treated with MSCs & RA group ((**H**) 400×). * Significant vs. Control group, # significant vs. Cis group, $ significant vs. Cis + MSCs group and ‡ significant vs. RA group.

Moreover, immunohistochemical examination for Bcl2 showed a significant increase in their expression levels in Cis group when compared with the control and vehicle groups ($p < 0.05$), while other groups showed significant increase in Bcl2, with the maximum significant rise in Cis + RA + MSCs group compared with Cis group ($p < 0.05$) (Figure 8A). The immunopositivity for Bcl2 appeared as cytoplasmic brown staining in hepatocytes at different times (days 3,7 and 11) in different groups (Cis, RA + Cis, MSCs + Cis and RA + MSCs + Cis groups) (Figure 8B–H).

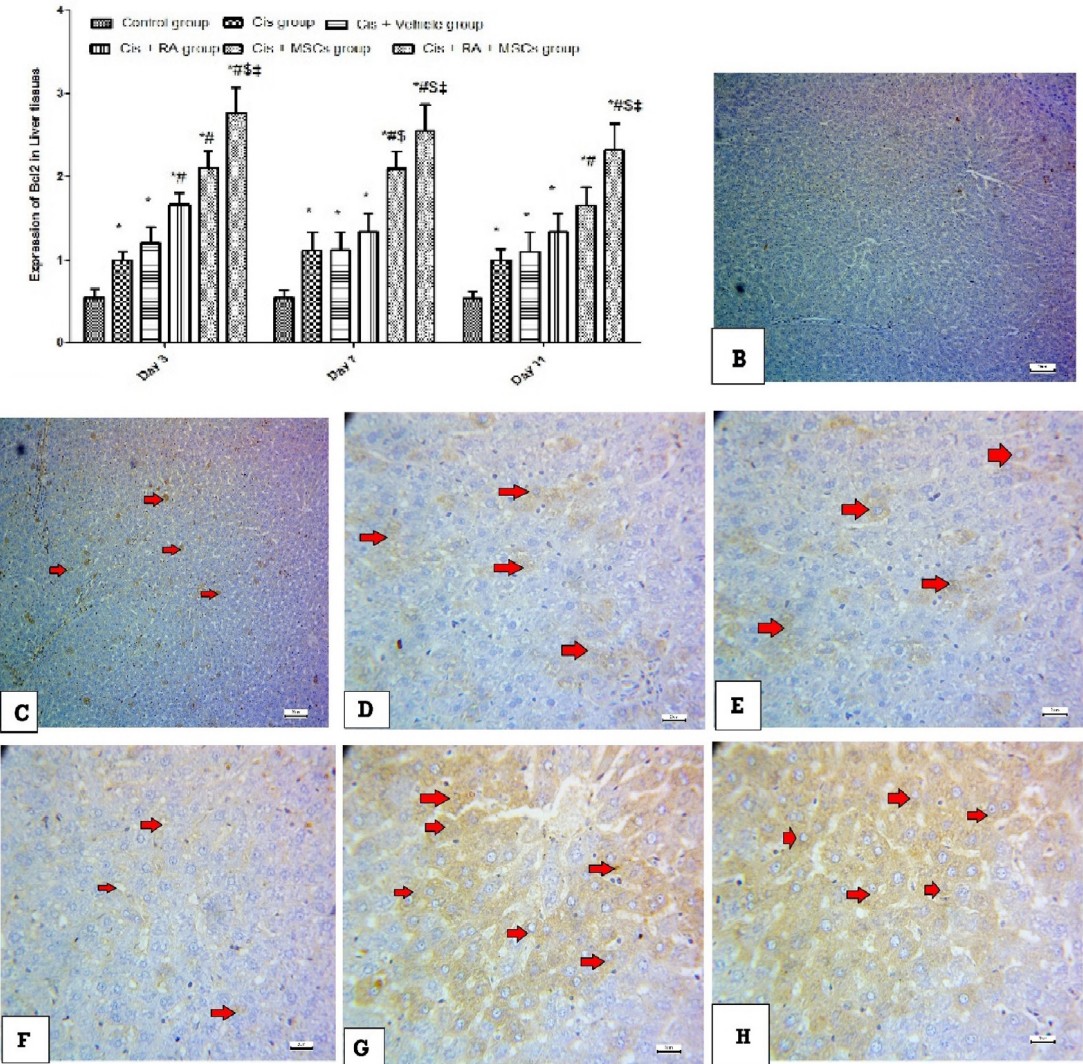

**Figure 8.** Immunohistopathological examination for the expression of Bcl2 in liver specimens. The score of Bcl2 expression in liver at different time intervals (3 days, 7 days and 11 days) showed significant rise in its expression in Cis group with significant rise in all treated groups (Cis + RA, Cis + BM-MSCs, Cis + RA+BM-MSCs groups) (**A**). Microscopic pictures of immunostained hepatic sections for Bcl2 from rats groups sacrificed at day 11 showing minimal staining in control group ((**B**) 100×), increased positive brown expression of Bcl2 (red arrows) in hepatocytes in Cis group

group at day 11 ((**C**) 100× and (**D**) 400×) and vehicle group ((**E**) 400×) and more upregulation Bcl2 expression in treated groups where positive cytoplasmic brown expression (red arrows) in hepatocytes is moderate in group treated with RA group ((**F**) 400×), MSCs group ((**G**) 400×), to marked expression in group treated with MSCs & RA group ((**H**) 400×). * Significant vs. Control group, # significant vs. Cis group, $ significant vs. Cis + MSCs group and ‡ significant vs. RA group.

### 3.6. Effect of BM-MSCs and RA and Combination of Both on Inflammatory Marker (IL-1Beta) in Cis-Induced Hepatotoxicity

Immunohistochemical examination for IL-1beta showed a significant increase in its expression level in Cis group when compared with the control and vehicle groups ($p <$ 0.05), while other groups showed significant decrease in IL-1beta with the maximum significant attenuation in Cis + RA + MSCs group compared with Cis group ($p < 0.05$) (Figure 9A). Immunopositivity for IL-1beta appeared as cytoplasmic brown staining in liver in different treated groups (RA + Cis, MSCs + Cis and RA + MSCs + Cis groups) at different times (days 3, 7 and 11) (Figure 9B–H).

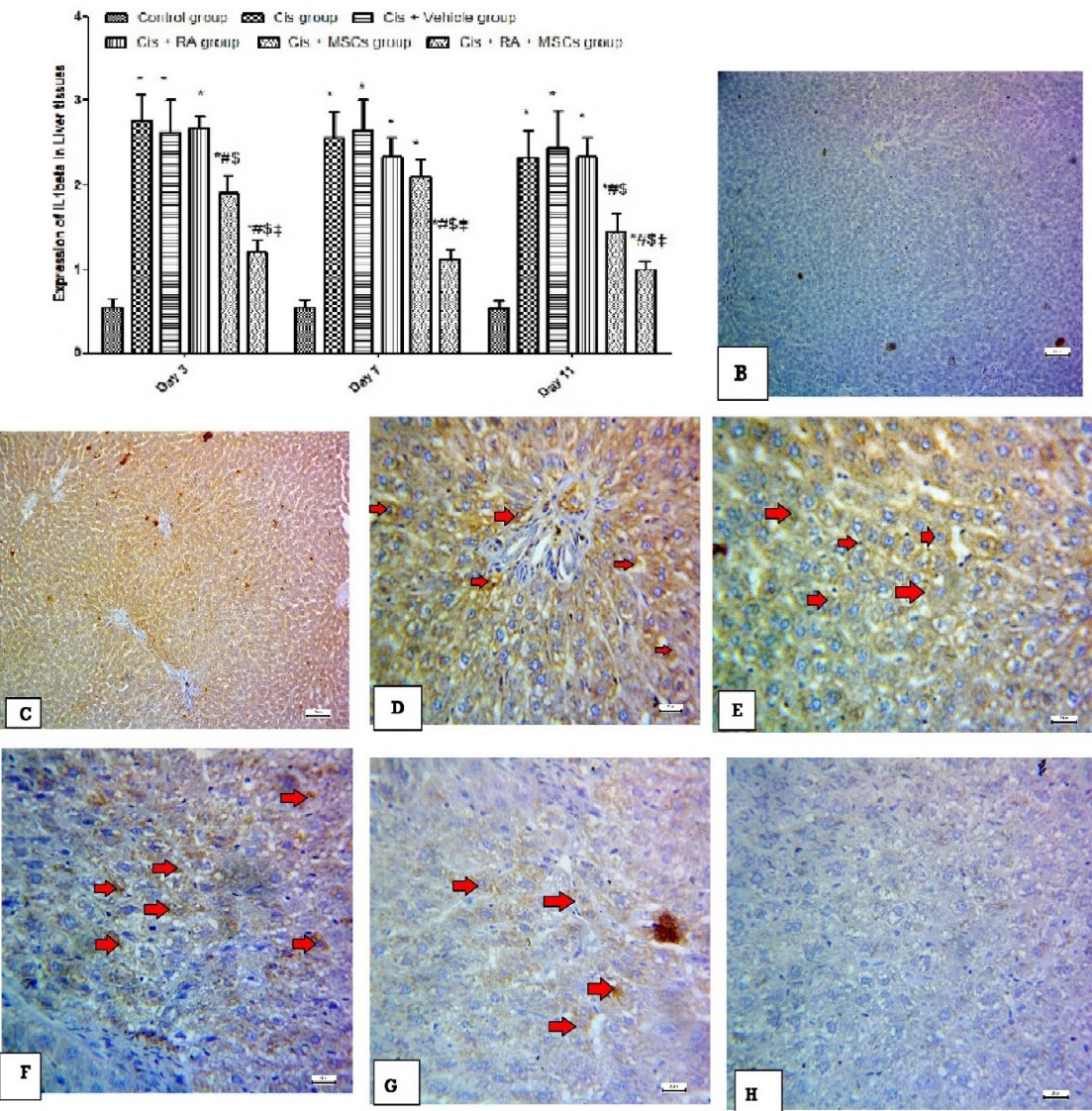

**Figure 9.** Immunohistopathological examination for the expression of IL-1beta in liver specimens. The score of Il1beta expression in liver at different time intervals (3 days, 7 days and 11 days)

showed significant rise in its expression in Cis group and significant attenuation in all treated groups (Cis + RA, Cis + BM-MSCs and Cis + RA + BM-MSCs groups) (**A**). Microscopic pictures of immunostained hepatic sections against Il1beta from rats groups sacrificed at day 11 showing negative staining in control group ((**B**) 100×), increased positive brown expression of IL-1beta (red arrows) in hepatocytes in Cis group group at day 11 ((**C**) 100× and (**D**) 400×) and vehicle group (**E**) in contrast to treated groups where positive cytoplasmic brown expression (red arrows) in hepatocytes is moderate in group treated with RA group ((**F**), MSCs group ((**G**) 400×), to minimal expression in group treated with MSCs & RA group ((**H**) 400×). * Significant vs. control group, # significant vs. Cis group, $ significant vs. Cis + MSCs group and ‡ significant vs. RA group.

## 4. Discussion

The main findings of the current study can be summarized as follows: (a) administration of Cis caused significant deteriorations on liver functions and morphology which was associated with enhanced oxidative stress, upregulation of NADPH oxidase, TNF-alpha and apoptotic protein 'caspase-3 in liver tissues; (b) treatment with either BM-MSCs or RA caused significant improvement in liver functions and morphology with significant attenuation in NADPH oxidase, TNF-alpha and caspase-3 expression; and (c) a combination of both RA and BM-MSCs offered more protective effect than each agent alone.

The current work demonstrated that administration of a single dose of Cis (7 mg/kg) i.p. results in significant liver cell damage and toxicity as evidenced by significant elevation in the serum levels of liver enzymes (ALT and AST). It has been concluded that the serum levels of ALT and AST are sensitive indicators for liver injury as liver cell injury results in an increase of its membrane permeability and causes leakage of the enzymes in plasma [33]. Moreover, we found that Cis administration results in significant deteriorations in liver morphology as shown by disturbed hepatocytes, inflammatory cell infiltrations, fatty changes and apoptosis in liver cells. These findings are in line with those reported by previous studies [2,4,5,34,35]. Although the mechanisms underlying the cis-induced hepatotoxicity are not well defined, in the current work we examined the role of oxidative stress, inflammation and apoptosis in this process. On the other hand, we found that pretreatment with either BM-MSCs and RA caused significant improvement in liver functions and morphology with the upper hand for BM-MSCs over RA. Moreover, treatment with BM-MSCs pretreated with RA offered a more powerful hepatoprotective effect than each agent did alone suggesting that RA potentiates the renoprotective effect of BM-MSCs against Cis-induced hepatoxicity.

The 2nd aim of the current study was to examine the role of oxidative stress in the process Cis-induced hepatotoxicity as well as in the possible hepatoprotective effects of BM-MSCs. The results of the current study confirmed an increase in oxidative stress in liver induced by Cis as evidenced by significant increase in MDA and decrease in hepatic antioxidants as GSH and catalase enzymes in Cis treated groups compared to normal control rats. These findings were in agreement with a previous study that confirmed Cis injection causes potential oxidative stress, which leads to cell death in liver tissue [36–38]. Interestingly, administration of BM-MSCs and RA combinations at different time intervals (3, 7 and 11 days) caused significant improvement in the markers of oxidative stress including MDA, GSH and catalase. We are suggesting an hepatoprotective effect through antioxidant effects of BM-MSCs in Cis-induced hepatotoxicity. In consistence with these findings, previous studies reported that BM-MSCs' early antioxidant activity could be a key role in creating an environment that promotes the proliferation of dedifferentiated epithelial cells that survive damage or resident stem cell activity, explains changes in MDA, GSH and CAT levels in the group treated with BM-MSCs [39] and RA. Moreover, we found the antioxidant effects of BM-MSCs pretreated with RA were more powerful than each agent did alone.

The next aim of the current work was to examine the expression of NADPH oxidase and pro-inflammatory cytokines TNF-$\alpha$ in Cis-induced hepatoxicity. These effects of

overexpression of the NADPH oxidase gene could be attributed to an increase in ROS production caused by increased $O_2$-production, culminating in lipid peroxidation [5]. Qiao, H. et al. [40] demonstrated that BM-MSCs have a positive effect on downregulated NADPH oxidase pathway which agrees with our findings. While downregulation of pro-inflammatory cytokines TNF-$\alpha$ may be due to chemotaxis of immune cells, and activation of additional cytokines exacerbates hepatic tissue inflammation [41]. This finding was in line with a previous study announced by Omar et al. [33], and proinflammatory cytokines such as TNF-$\alpha$ and IL-6 are released in response to oxidative stress [6,42] which accelerate the process of apoptosis and cellular damage. Our results showed that the BMSCs and RA combination offers strong anti-inflammatory protection against Cis-induced inflammation, owing to a reduction in the proinflammatory TNF-$\alpha$. There is a significantly greater anti-inflammatory effect of the BM-MSCs and RA combination than of the BM-MSCs group or RA group when compared to the Cis group. As was proposed, MSCs could reduce the influx of inflammatory cells from the bone marrow [43], and RA is well-known for its anti-inflammatory properties [44,45]. These effects appear to be mediated by a decrease in Th1 cytokines such as TNF-$\alpha$ [46,47], explaining the downregulation of TNF-$\alpha$.

According to immunohistochemistry, this study showed that Cis induced apoptotic cell death in hepatic tissue which determined by the main apoptotic caspase: caspase-3 expression in liver tissue. This study revealed that there is a substantial increase in the expression of the caspase-3 (a marker of apoptosis) in liver tissues in the Cis group. These findings are in harmony with the findings of Eckle et al. [48]. Our findings show that a combination of BM-MSCs and RA causes inflammatory cytokine modulation and apoptosis suppression at Cis-induced hepatotoxicity, which could be attributed to an increase in oxidative stress-mediated apoptosis. Moreover, Berardis et al. [49] demonstrated that BM-MSCs has anti-apoptotic and pro-mitotic effects on cultured hepatocytes. This study sheds light on the fact that MSCs deserve substantial attention as a promising approach for regenerative medicine concerning liver disease, as they are regarded as a safe and potentially effective therapeutic strategy for patients suffering from chronic liver disease [50]. These findings might be due to the fact that RA is a good inducing factor that helps cell differentiation via upregulation of some genes that influence the cell cycle and differentiation [51]. Moreover, RA has already been shown to exert antiapoptotic and antioxidative activity in various cells [52]. Therefore, the findings of the current study confirmed the previous findings that BM-MSC and RA could have a synergistic effect with each other to ameliorate Cis-induced hepatotoxicity.

## 5. Conclusions

Our study is the first study to suggest a combination of BM-MSCs and RA in Cis-induced hepatotoxicity. Our experiment found that administration of a combination of BM-MSCs and RA ameliorates Cis-induced hepatotoxicity more than each agent did alone due to attenuation of oxidative stress, apoptotic protein (caspase-3), inflammatory cytokine TNF-$\alpha$ and NADPH oxidase expression in liver tissues. Therefore, a BM-MSCs and RA combination may be recommended for clinical use, by adding it to the chemotherapy-based treatment of cancer patients to delay the adverse effects induced by hepatotoxicity caused by the treatment.

**Author Contributions:** Conceptualization, M.M.A., A.M.H., B.H.M., and N.A.H.; methodology, M.M.A., B.H.M., N.M.B. and M.M.M.K.; formal analysis, M.M.A., A.M.H. and B.H.M.; investigation, M.M.A., A.M.H. and B.H.M.; data curation, M.M.A., A.M.H. and B.H.M., N.A.H.; writing—original draft preparation, M.M.A., A.M.H. and B.H.M., N.A.H.; writing—review and editing, A.M.H., B.H.M. and N.A.H.; supervision, M.M.A., A.M.H., B.H.M., and N.A.H. All authors have read and agreed to the published version of the manuscript.

**Funding:** This research received no external funding.

**Institutional Review Board Statement:** The study was conducted in accordance with the Declaration of Helsinki, and approved by the Research Ethics Committee, Faculty of Veterinary Medicine, Mansoura University, Mansoura, Egypt), Code No. ph.D/17.

**Informed Consent Statement:** Not applicable.

**Data Availability Statement:** All raw data are available on reques.

**Acknowledgments:** We acknowledge Walaa Awaden for helping us in histopathological slides preparation.

**Conflicts of Interest:** The authors declare no conflicts of interest.

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
