# Peer review of "Retinoic Acid Potentiates the Therapeutic Efficiency of Bone Marrow-Derived Mesenchymal Stem Cells (BM-MSCs) against Cisplatin-Induced Hepatotoxicity in Rats"

_scipharm, doi:10.3390/scipharm90040058_

Round 1

Reviewer 1 Report

Authors here demonstrate the hepato-protective effects of  BM-MSCs and RA against Cisplation toxicity through their antioxidant, anti-inflammatory, and anti-apoptotic properties. The manuscript as whole is well written and has some interesting results. I have few suggestions to increase the scientific rigor of the experimental design here and to strengthen the claims made in this manuscript. The manuscript may be accepted following major revision.

Comments,

  1. Please provide references for the dose selection of BM-MSCs and RA (1mg/kg).
  2. Please provide more details for the scores of hepatic lesions (L161) Please refer to Front. Pharmacol., 10 November 2020 | https://doi.org/10.3389/fphar.2020.563750; PLoS ONE 11(3): e0151649
  3. It would be interesting to see the expression of p53, levels of Bcl-2 to check the antiapoptotic effect of BM-MSCs and RA against Cis-induced hepatotoxicity.
  4. Another important assay would be to check the levels of proinflammatory cytokines IL-10, IL-6, IL-1β to assess the anti-inflammatory property of BM-MSCs and RA against Cis-induced hepatotoxicity.

Author Response

Reviewer’s # 1

Authors here demonstrate the hepato-protective effects of BM-MSCs and RA against Cisplatin toxicity through their antioxidant, anti-inflammatory, and anti-apoptotic properties. The manuscript as whole is well written and has some interesting results. I have few suggestions to increase the scientific rigor of the experimental design here and to strengthen the claims made in this manuscript. The manuscript may be accepted following major revision.

Comments,

Comment

  1. Please provide references for the dose selection of BM-MSCs and RA (1mg/kg).

Response

The reference for dosing of RA and BM-MSCs was added to the revised manuscript

Comment

  1. Please provide more details for the scores of hepatic lesions (L161) Please refer to Front. Pharmacol., 10 November 2020 | https://doi.org/10.3389/fphar.2020.563750; PLoS ONE 11(3): e0151649

Response

We would like to thank the reviewer for this comment. The above-mentioned reference used a scoring system for hepatic lesion in chronic active hepatitis in human not in animals and the scoring system. We added more details about the scoring system of histological changes from reference # 30 with maximum score of 12.

Comment

  1. It would be interesting to see the expression of p53, levels of Bcl-2 to check the antiapoptotic effect of BM-MSCs and RA against Cis-induced hepatotoxicity.

Response

Immunostaining for p53 and Bcl2 expression by immunostaining was added to revised version of the manuscript

Comment

  1. Another important assay would be to check the levels of proinflammatory cytokines IL-10, IL-6, IL-1β to assess the anti-inflammatory property of BM-MSCs and RA against Cis-induced hepatotoxicity.

Response

We the expression of IL-1beta by immunostaining in liver tissues was added in the revised manuscript

Reviewer 2 Report

The study by Azam MMME et al examines the potential for addition of retinoic acid to bone marrow mesenchymal stem cells in protecting against cisplatin-induced liver injury in rats. Though the authors have carried out several experiments, issues with clarity and consistency make interpretation of the data problematic. Additionally, there is a lack of clinical relevance for the study and several very important controls are missing as well.

Major points

1) Several important control groups seem to be missing from the study- namely rats given BM-MSC or retinoic acid alone, without cisplatin. Since the MSC were injected in culture media, an additional control (Cisplatin+culture media alone) is also missing. These are critical to interpret the data.

2) The introduction contains no proper justification for carrying out these studies. It seems to be merely a collection of paragraphs discussing cisplatin toxicity, bone marrow MSC and retinoic acid, without any connection between them, or a rationale for why it was tested. For example, what percentage of patients on cisplatin get liver injury? Which pathways of cisplatin toxicity in the liver have MSC shown to target? How do the authors align their study with the fact that retinoic acid has in fact been shown to potentiate the cytotoxic action of cisplatin (PMID: 23867314)? None of these pertinent issues are discussed.

3) This is all the more important since cisplatin induced hepatotoxicity is rare at standard doses and transient even at higher doses in humans. So why is it clinically relevant to study this? What effect does the BM-MSC+retinoic acid at this dose have on nephrotoxicity due to cisplatin, which seems to be more clinically relevant?

4) What was the justification for choice of cisplatin dose in the study? How does this compare to what is given to humans?

5) How were the bone-marrow MSC characterized? What assays were run to confirm that the cell population isolated was indeed MSC? No information on this very important point is provided.

6) The time course data for all parameters should be presented as either a line graph or bar graph rather than tables, which would significantly improve understanding of the data.

7) What are the units for measurement of MDA, GSH and Catalase activity? This needs to be included.

8) The gene expression data is difficult to interpret due to the significant variation. The authors should present the data as a scatter plot, which would help determine whether a few data points are skewing the results.

9) The images showing liver histology need to be repeated since the current images are of very poor quality and cannot be readily interpreted. ALT values of around 300 IU/L would indicate some liver necrosis, which seems to be not evident in the provided images. Lower magnification images should be provided to confirm that the damage was widespread and not localized to small regions.

10) The caspase 3 immunostaining also does not seem to be accurate since the signals seem to be non-specific staining independent of cellular location and do not always seem within hepatocytes. In addition, the images under higher magnification show no morphological signs of apoptosis in cells which seem to have high caspase 3 signal.  

Minor points

1) Page 3, line 102- ….”soaked for two minutes in a 70% (vol/vol) solution”. A solution of what? The sentence implies that the whole rats were soaked. Is this correct, or only the limbs?

2) Page 4, line 175- Leica DFC280 seems to be the camera, and it would be useful to also provide detail of the type of microscope used.

Author Response

Reviewer’s # 2

The study by Azam MMME et al examines the potential for addition of retinoic acid to bone marrow mesenchymal stem cells in protecting against cisplatin-induced liver injury in rats. Though the authors have carried out several experiments, issues with clarity and consistency make interpretation of the data problematic. Additionally, there is a lack of clinical relevance for the study and several very important controls are missing as well.

Major points

Comment

1) Several important control groups seem to be missing from the study- namely rats given BM-MSC or retinoic acid alone, without cisplatin. Since the MSC were injected in culture media, an additional control (Cisplatin+culture media alone) is also missing. These are critical to interpret the data.

Response

A group of cisplatin and vehicle or culture media was added to the revised manuscript

Comment

2) The introduction contains no proper justification for carrying out these studies. It seems to be merely a collection of paragraphs discussing cisplatin toxicity, bone marrow MSC and retinoic acid, without any connection between them, or a rationale for why it was tested. For example, what percentage of patients on cisplatin get liver injury? Which pathways of cisplatin toxicity in the liver have MSC shown to target? How do the authors align their study with the fact that retinoic acid has in fact been shown to potentiate the cytotoxic action of cisplatin (PMID: 23867314)? None of these pertinent issues are discussed.

Response

Few sentences were added to the introduction of revised manuscript to clarify all requested points and the following references were added

  1. Abd Rashid, N., Abd Halim, S. A. S., Teoh, S. L., Budin, S. B., Hussan, F., Ridzuan, N. R. A., & Jalil, N. A. A. (2021). The role of natural antioxidants in cisplatin-induced hepatotoxicity. Biomedicine & Pharmacotherapy144, 112328.‏
  2. Sarhan, M., El Serougy, H., Hussein, A. M., El-Dosoky, M., Sobh, M. A., Fouad, S. A., ... & Elhusseini, F. (2014). Impact of bone-marrow-derived mesenchymal stem cells on adriamycin-induced chronic nephropathy. Canadian journal of physiology and pharmacology92(9), 733-743.‏

Comment

How do the authors align their study with the fact that retinoic acid has in fact been shown to potentiate the cytotoxic action of cisplatin (PMID: 23867314)?

Response

This study was invitro not in vivo study which concluded that ATRA could significantly improve the effect of cisplatin, which is at least partially attributed to ATRA-induced differentiation of HCC TICs, and the subsequent decrease in this chemo-resistant subpopulation, however in the current study we investigated the effect of BM-MSCs-which are pretreated or preconditioned with ATRA in culture before using it for treated. We think there is no controversy between this study and our work as we postulated that ATRA could enhance differentiation of BM-MSCs hence improve its therapeutic efficacy

Comment

3) This is all the more important since cisplatin induced hepatotoxicity is rare at standard doses and transient even at higher doses in humans. So why is it clinically relevant to study this? What effect does the BM-MSC+retinoic acid at this dose have on nephrotoxicity due to cisplatin, which seems to be more clinically relevant?

Response

The dose of BM-MSCs given in the current study was 1x106 i.e. one million cells which seems to be effective in a previous study by Khedr et al. 2022 against cisplatin induced nephrotoxicity. Also, previous clinical studies used the same dose of BM-MSCs i.e. 1x106 such as Kantarcioglu et al. 2012 (they administrated stem cells via peripheral veins in liver cirrhosis) and El-Ansary et al. 2012 (they administrated stem cells via intrasplenic vein and peripheral veins in patients with chronic liver disease)

  1. Khedr, M., Barakat, N., El-Deen, I. M., & Zahran, F. (2022). Impact of preconditioning stem cells with all-trans retinoic acid signaling pathway on cisplatin-induced nephrotoxicity by down-regulation of TGFβ1, IL-6, and caspase-3 and up-regulation of HIF1α and VEGF. Saudi Journal of Biological Sciences29(2), 831-839.‏
  2. Kantarcioglu M. Efficacy of Exvivo Expanded Autologous Mesenchymal Stem Cell Transplantation Via Peripheral Vein in Patients With Liver Cirrhosis. clinicaltrials.gov; 2012. Available from: https://clinicaltrials.gov/ct2/show/NCT01499459.
  3. El-Ansary M, Abdel-Aziz I, Mogawer S, et al. Phase II trial: undifferentiated versus differentiated autologous mesenchymal stem cells transplantation in Egyptian patients with HCV induced liver cirrhosis. Stem Cell Rev Rep 2012;8:972-81

Comment

4) What was the justification for choice of cisplatin dose in the study? How does this compare to what is given to humans?

Response

In previous studies we found 6 doses for cisplatin to induce hepatoxicity in rats , 6 mg/kg (Khedr et al., 2022), 7 mg/kg (Cüre et al., 2016) and 10 mg/Kg (Taghizadeh et al., 2021), so we did a pilot study in which we used 3 doses 6 mg/Kg, 7 mg/Kg and 10 mg /Kg and found 7 mg /kg produced significant impairment in liver tissues without great loss in rats as 10 mg/Kg produces 90% mortality in rats.

  • Cüre MC, Cüre E, Kalkan Y, KırbaÅŸ A, Tümkaya L, Yılmaz A, Türkyılmaz AK, ÅžehitoÄŸlu İ, Yüce S. Infliximab Modulates Cisplatin-Induced Hepatotoxicity in Rats. Balkan Med J. 2016 Sep;33(5):504-511. doi: 10.5152/balkanmedj.2016.150576. Epub 2016 Sep 1. PMID: 27761277; PMCID: PMC5056652.
  • Taghizadeh F, Hosseinimehr SJ, Zargari M, Karimpour Malekshah A, Mirzaei M, Talebpour Amiri F. Alleviation of cisplatin-induced hepatotoxicity by gliclazide: Involvement of oxidative stress and caspase-3 activity. Pharmacol Res Perspect. 2021 May;9(3):e00788. doi: 10.1002/prp2.788. PMID: 34003600; PMCID: PMC8130655.

Comment

5) How were the bone-marrow MSC characterized? What assays were run to confirm that the cell population isolated was indeed MSC? No information on this very important point is provided.

Response

Characterization of BM-MSCs was added to the revised manuscript

Comment

6) The time course data for all parameters should be presented as either a line graph or bar graph rather than tables, which would significantly improve understanding of the data.

Response

All data were represented in graph form

Comment

7) What are the units for measurement of MDA, GSH and Catalase activity? This needs to be included.

Response

The units for measurement of these markers were added to revised manuscript

Comment

8) The gene expression data is difficult to interpret due to the significant variation. The authors should present the data as a scatter plot, which would help determine whether a few data points are skewing the results.

Response

New clear graphs for the gene expression results were made in the revised manuscript after adding the new group.

Comment

9) The images showing liver histology need to be repeated since the current images are of very poor quality and cannot be readily interpreted. ALT values of around 300 IU/L would indicate some liver necrosis, which seems to be not evident in the provided images. Lower magnification images should be provided to confirm that the damage was widespread and not localized to small regions.

Response

New images for the histopathological changes including lower magnification images were added to the revised manuscript

Comment

10) The caspase 3 immunostaining also does not seem to be accurate since the signals seem to be non-specific staining independent of cellular location and do not always seem within hepatocytes. In addition, the images under higher magnification show no morphological signs of apoptosis in cells which seem to have high caspase 3 signal.  

Response

New images for caspase-3 were added to the revised manuscript

Comment

Minor points

1) Page 3, line 102- ….”soaked for two minutes in a 70% (vol/vol) solution”. A solution of what? The sentence implies that the whole rats were soaked. Is this correct, or only the limbs?

Response

Great thanks for this comment. This typing error was corrected in revised manuscript

Comment

2) Page 4, line 175- Leica DFC280 seems to be the camera, and it would be useful to also provide detail of the type of microscope used.

Response

Great thanks the microscope and its version was added to the revised version.

Round 2

Reviewer 1 Report

Manuscript may be accepted

Author Response

Reviewer’s # 1

Comment

Manuscript may be accepted

Response

Great thanks you
